# ROMP and Vinyl Polynorbornenes with Vanadium(III) and Nickel(II) diNHC Complexes

**DOI:** 10.3390/ijms26146691

**Published:** 2025-07-12

**Authors:** Katarzyna Halikowska-Tarasek, Elwira Bisz, Dawid Siodłak, Błażej Dziuk, Wioletta Ochędzan-Siodłak

**Affiliations:** 1Department of Chemistry and Pharmacy, Opole University, Oleska 48, 45-052 Opole, Poland; katarzyna.halikowska-tarasek@uni.opole.pl (K.H.-T.); ebisz@uni.opole.pl (E.B.); dsiodlak@uni.opole.pl (D.S.); 2Department of Chemistry, Wroclaw University of Science and Technology, Norwida 4/6, 50-373 Wroclaw, Poland; blazej.dziuk@pwr.edu.pl

**Keywords:** norbornene polymerization, vinyl-addition polymerization, ring-opening metathesis polymerization, *N*-heterocyclic carbene (NHC) ligands, nickel(II) complexes, vanadium(III) complexes

## Abstract

The polymerization of norbornene can occur via ring-opening metathesis polymerization (ROMP) or vinyl-addition pathways, each yielding polynorbornene with distinct structures and properties. This study reports on the synthesis and catalytic application of a new class of vanadium(III) and nickel(II) complexes bearing *N*-heterocyclic carbene ligands, based on the IPr* framework, for the polymerization of norbornene. The vanadium(III) complexes, activated by diethylaluminum chloride and in the presence of ethyl trichloroacetate, showed activity in ROMP. In contrast, the nickel(II) complexes, activated by methylaluminoxane, exhibited catalytic activity toward vinyl-addition polymerization. Characterization by GPC, NMR, and FTIR confirmed the formation of both ring-opening metathesis polymerization and vinyl-type-derived polynorbornenes, with vinyl-type polymers showing significantly higher molecular weights. Structural variations in the *N*-heterocyclic carbene ligands, particularly the linker length between imidazole donors, were found to strongly influence polymer molecular weight and the morphology of polynorbornenes.

## 1. Introduction

The structure and characteristics of polynorbornene (PNB) synthesized using transition metal-based catalysts are influenced by the polymerization mechanism. Norbornene can undergo polymerization through two fundamental pathways, determined by the used catalyst, as follows: ring-opening metathesis polymerization (ROMP) and vinyl polymerization (Figure 1) [1,2,3,4,5,6,7,8,9,10,11,12,13,14,15,16,17,18,19,20,21]. Each polymerization pathway results in a distinct type of polynorbornene with unique microstructural features and properties. Polymers synthesized via the ROMP mechanism are unsaturated and typically demonstrate good solubility in a range of solvents and a low glass transition temperature (Tg ≈ 35 °C). The flexible backbone with reactive double bonds causes thermal and oxidative instability. On the other hand, reactive double bonds allow for a variety of functionalization, making poly(norbornene) ROMP suitable for optical components, biomedical scaffolds, hydrogels, drug delivery systems, and stimuli-responsive materials [4,5,6,7,8,9,10,11,12]. In contrast, the vinyl polymerization of norbornene produces saturated, 2,3-inserted polymers, which possess unique chemical and physical properties, such as high thermal stability, and high glass transition temperature values ranging from 180 to 370 °C, depending on the functional groups. It is amorphous, optically transparent, chemically resistant, and has low moisture absorption, making it ideal for high-performance applications such as microelectronic dielectrics, optical films, and LCD cover layers [1,2,3,12,13,14,15,16,17,18,19,20,21].

From a practical standpoint, enhancing catalytic activity and precisely controlling product structures are key objectives in the development of polymerization catalysts. These goals can be achieved by selecting the metal center in the complex and modifying its coordination sphere with different ligands. *N*-Heterocyclic carbene (NHC) ligands are a class of compounds that have been extensively studied and widely applied in organometallic chemistry and are now commonly used as alternatives to phosphines [22,23,24,25,26,27]. They are frequently used to stabilize transition metal complexes and have found broad applications, particularly in homogeneous catalysis [28,29,30,31,32,33].

Our research group has been interested in the synthesis and catalytic application of transition metal complexes with *N*-heterocyclic carbene ligands [34] in technologically relevant reactions like olefin polymerization [35,36,37,38,39,40,41,42,43,44,45,46,47,48]. In particular, we have recently started a systematic study aimed at evaluating the reactivity of chelating dicarbene vanadium(III) and nickel(II) complexes based on the well-known IPr* framework (IPr* = (2,6-bis(diphenylmethyl)-4-methylphenyl)imidazol-2-ylidene) [49,50,51] and the bridging group between the *N*-heterocyclic carbene donors.

While metal *N*-heterocyclic carbene complexes have been applied in olefin polymerizations, for norbornene homopolymerization, the study concerns monodentate NHC ligands, or a bidentate with other functionalities, mainly using nickel as metal center [47]. The application of NHC vanadium complexes is not much explored and also concerns the monodentate carbene ligand [48]. Applications of bidentate NHC ligands in catalysts for olefin addition polymerization remain relatively limited [39,42,43,44]. To our knowledge, the present study is the first application of this type of bidentate NHC ligands in the V(III)- and Ni(II)-catalyzed homopolymerization of norbornene.

In this study, we focused on the synthesis and characterization of polynorbornenes obtained using different catalysts. In this context, we synthesized a new class of dihalo—vanadium(III) complexes (diNHC)VCl_3_ and nickel(II) complexes (diNHC)NiCl_2_ (Figure 2), based on novel chelating bidentate bis(imidazol-2-ylidene) ligands—and evaluated their catalytic efficiency for norbornene homopolymerization in the presence of diethylaluminium chloride (AlEt_2_Cl) or methylaluminoxane (MMAO), respectively, as cocatalyst. The resulting polynorbornenes were characterized by gel permeation chromatography (GPC), ^1^H and ^13^C NMR, and FTIR and SEM.

## 2. Results

### 2.1. Synthesis of the diNHC Pre-Ligands

The bidentate pre-ligands **2a** and **2b** were synthesized following our previously published synthetic route [34]. Notably, pre-ligand **2a** was synthesized and fully characterized for the first time by our research group; the complete characterization is provided in the Appendix A. Both salts were obtained by the reaction of 2 equivalents of imidazole with 1 equivalent of an alkyl halide. In our previous study, dihaloalkanes reacted with an excess of imidazolium salt in acetonitrile to enhance bridged bis(imidazolium) salts. Acetonitrile was chosen as a solvent instead of the more commonly used THF because it can be used at higher temperatures, thus decreasing reaction time. The white/off-white bis(imidazolium) salts were obtained in good yields (Figure 3).

### 2.2. Synthesis of Vanadium(III) Complexes

Following the modified procedures proposed by Danopoulos [52], Theopold [53], and Radius [54], we synthesized vanadium(III) complexes **3a** and **3b** by using readily available VCl_3_(THF)_3_ with **2a** and **2b** leading to the formation of [VCl_3_(IPr*^diNHC^)_2_] complexes (Figure 4).

The resulting vanadium(III) complexes were obtained as light brown solids with good yields of 50% and 71%, respectively. These air- and moisture-sensitive compounds are well soluble in solvents like CH_2_Cl_2_ at room temperature. The structural characterization of **3a** and **3b** was performed using FTIR spectroscopy and mass spectrometry (MS). The MS spectra (Appendix A) confirmed the predicted mass of the [V(IPr*^diNHC^)_2_]^3+^ ions, showing peak *m*/*z* of 367.1862 for [M]^3+^ (**3a**) and 382.1798 for [M+3H]^3+^ (**3b**). Due to the paramagnetic nature of these vanadium(III) complexes, NMR spectroscopy did not provide useful data.

### 2.3. Norbornene Polymerization Catalyzed by Vanadium(III) Complexes

The polymerization of norbornene was investigated with **3a** and **3b** under identical experimental conditions (Table 1). Methylaluminoxane (MMAO) was not employed due to its complex and poorly defined structure, which can lead to the unpredictable interactions and uncontrolled activation of vanadium species [55,56]. Instead, diethylaluminum chloride (AlEt_2_Cl) was used to achieve more controlled and efficient activation. Upon activation with diethylaluminum chloride (AlEt_2_Cl) and in the presence of ethyl trichloroacetate (ETA), both complexes exhibited good activity in norbornene polymerization. The polymerization reaction time of 2 h was selected based on conducting the reaction over various durations, i.e., 2, 24, 48, 72, and 96 h. However, the reactions carried out for 2 h gave the highest yield (1.76–1.86 g). The catalytic activity was, respectively, 293.3 × 10^3^ and 310.0 × 10^3^ g_PNB_/mol_Mt_/h for vanadium(III) complexes **3a** and **3b**. The microstructure of polynorbornenes was characterized by NMR (Figure 5) and IR (Figure 6). All polynorbornenes obtained from these catalytic systems gave similar spectroscopic characteristics, and thus, further study was pursued with vanadium(III) catalyst **3b**.

To clearly determine the structure of the obtained polymers, both ^1^H and ^13^C NMR spectra were analyzed, as also found in the literature [6,12,57]. In the ^1^H NMR spectra, the characteristic peaks around 5.8–5.6 ppm (Figure 5A and Appendix A) are assigned to the protons of double bonds. The ratio of cis and trans double bonds for polynorbornene is roughly 1:3. The ^13^C NMR spectra (Figure 5B and Appendix A) exhibit four distinct groups of resonance peaks and the resonances for olefinic carbons in cis double bonds are found around 135.9 ppm and for trans double bonds around 134.5 ppm.

Furthermore, in the FTIR spectra (Figure 6 and Appendix A), C-H stretching vibration absorption of trans and cis double bonds appeared around 951cm^−1^ and 748 cm^−1^, respectively. Additionally, absorption bands are observed in the 1605–1723 cm^−1^ region, and 951 cm^−1^, which are typically associated with the stretching vibrations of trans C=C double bonds in the ROMP polynorbornene [12,17,18].

The melting temperature (T_m_) of polynorbornene obtained with vanadium(III) catalyst **3b** was 274 °C (Appendix A), and for polynorbornene obtained with vanadium(III) catalyst **3a** it cannot be determined. The molecular weight of the polynorbornenes obtained with vanadium(III) catalysts **3a** and **3b** was found to be low (Appendix A). The weight-average molecular weight (M_w_) determined by GPC was 2.03 and 2.10 kDa, respectively. The molecular weight distribution, regardless of the type of catalyst used, was very narrow and amounted to 1.3.

Based on those analyses, we can conclude that norbornene polymerization with these catalytic systems occurs via a ROMP mechanism. These polymers exhibit a very narrow molecular weight distribution (low dispersity), which is indicative of a high degree of polymer chain uniformity and proves a well-controlled polymerization process.

### 2.4. Synthesis of Nickel(II) Complexes

Following the modified procedures proposed by Danopoulos [52], Stieler [58], and Kilyanek [59], we synthesized nickel(II) complexes **4a** and **4b** by using NiCl_2_(DME) with **2a** and **2b**, leading to the formation of [NiCl_2_(IPr*^diNHC^)_2_] complexes (Figure 7).

The resulting nickel(II) complexes **4a** and **4b** were obtained as light green solids, with good yields of 50% and 60%, respectively, as before. These air- and moisture-sensitive compounds are well soluble in solvents like CH_2_Cl_2_ at room temperature. The structural characterization of **4a** and **4b** was performed using FTIR spectroscopy and mass spectrometry (MS). The MS spectra (Appendix A) confirmed the predicted mass of the [Ni(IPr*^diNHC^)_2_]^2+^ ions, showing peaks *m*/*z* of 563.1896 for [M+2H]^2+^ (**4a**) and 575.3338 for [M]^2+^ (**4b**). Due to the paramagnetic nature of these nickel(II) complexes, NMR spectroscopy did not provide useful data.

### 2.5. Norbornene Polymerization Catalyzed by Nickel(II) Complexes

The polymerization of norbornene was investigated with nickel(II) catalysts **4a** and **4b** under identical experimental conditions (Table 1). AlEt_2_Cl was found to be ineffective in activating the nickel(II) complexes toward the polymerization of norbornene. This lack of activity is likely due to insufficient alkylation or the incomplete generation of the catalytically active species under these conditions [40]. In contrast, upon activation with MMAO, both nickel(II) complexes exhibited good catalytic activity in the polymerization of norbornene. The polymerization reaction time of 2 h was selected based on conducting the reaction over various durations (2, 24, 48, 72, and 96 h), which were characterized by the yield (0.39–0.55 g). The catalytic activity was, respectively, 91.7 × 10^3^ and 65.0 × 10^3^ g_PNB_/mol_Mt_/h for complexes **4a** and **4b**. The microstructure of polynorbornenes were characterized by NMR (Figure 8) and FTIR (Figure 9). All polynorbornenes obtained from these catalytic systems gave similar spectroscopic characteristics, and thus, the remainder of the study was pursued with nickel(II) catalyst **4b**. In ^1^H NMR spectra, the absence of olefin signals, indicated by the lack of resonance at 5.00–6.00 ppm (Figure 8A), where double bonds typically appear in metathesis-type polynorbornene, confirms that the obtained spectra correspond to vinyl-type polynorbornene [17,18]. The ^13^C NMR spectra (Figure 8B) exhibit four distinct groups of resonance peaks. Based on data from the literature, these peaks correspond to methylene and methine carbon signals characteristic of vinyl-type addition polymers. Specifically, the resonances at 29.8–31.4 ppm are assigned to the C5/C6 carbons, 34.9–37.3 ppm to the C7 carbon, 37.4–39.8 ppm to the C1/C4 carbons, and 46.9–52.6 ppm to the C2/C3 carbons [17,18,19,20].

Furthermore, the FTIR spectra of polynorbornene (Figure 9) exhibit a characteristic absorption peak around 939 cm^−1^, which corresponds to the norbornene ring in vinyl-type addition polymers. Additionally, no absorption bands are observed in the 1620–1680 cm^−1^ region, nor around 960 cm^−1^, which are typically associated with the stretching vibrations of trans C=C double bonds in the ROMP polynorbornene [12,17,18].

The glass transition temperature (Tg) of polynorbornenes obtained with nickel(II) catalyst **4a** was 124 °C, and with **4b** it was 152 °C. The thermogravimetric analysis (TGA) of these polymers demonstrates their high thermal stability, which the PNB, in particular, obtained using the catalyst with the longer C4 bridge (**4b**) (Appendix A). The average molecular weights and dispersity of polynorbornenes obtained with catalysts **4a** and **4b** were higher than those obtained for polymers obtained with vanadium catalysts (Appendix A). The weight-average molecular weight (Mw) determined by GPC was 73.14 and 97.23 kDa, respectively. Moreover, with the extension of the linker in the ligand, the average molecular weight increased, and the polymer dispersity decreased (Mw/Mn = 2.4 and 1.9, respectively). Based on these analyses, we can conclude that the polymerization of norbornene with these catalytic systems proceeds via the vinyl coordination mechanism.

Moreover, scanning electron microscope (SEM) imaging of the polynorbornenes obtained using nickel(II) complexes with varying linker lengths between imidazole donors revealed a strong influence on polymer morphology (Figure 10, Appendix A). The polynorbornene synthetized using catalyst **4a** is in the form of irregular, aggregated fine particles (Figure 10A). In contrast, the polynorbornene obtained with catalyst **4b** displays a morphology dominated by elongated, fibrous, and layered structures (Figure 10B). The PNB obtained over the nickel catalyst with the pre-ligand 2b (C4 bridge) has a smooth surface without visible defects. Compared to the **4a**-derived polynorbornene, the **4b**-derived sample shows more coherent and compatible morphology, suggesting advanced and improved polymer packing. These observations indicate that linker length has a significant impact on the resulting polymer morphology.

## 3. Materials and Methods

### 3.1. Synthesis

All reactions were carried out under an inert (argon) atmosphere using the Schlenk technique and glovebox. Toluene and THF were dried over sodium/benzophenone and distilled under nitrogen prior to use. Norbornene (99%) and dichloromethane were purchased from Thermo Scientific, MMAO-12; AlEt_2_Cl and VCl_3_(THF)_3_ were purchased from Sigma-Aldrich; and NiCl_2_(DME) was purchased from AmBeed. These were deoxygenated prior to use. The pre-ligands **2a**–**2b** [34] were synthesized according to the literature, and complexes **3a**–**3b** [52,53,54] and **4a**–**4b** [52,58,59] were synthesized according to modified literature procedures. The detailed synthetic procedure and all obtained results (NMR, MS, FTIR, DSC, TGA, GPC, and SEM) are presented in the Appendix A.

### 3.2. Methods

^1^H NMR and ^13^C NMR spectra were recorded on a Bruker spectrometer Bruker (Corporation, Billerica, MA, USA) at 400 (^1^H NMR) and 100 MHz (^13^C NMR). The spectra were recorded in CDCl_3_ (contains 0.05% (*v*/*v*) TMS) at room temperature.

Mass spectroscopy (MS) was performed on a Waters Xevo G3 Q-TOF instrument (Waters, Milford, MA, USA).

Fourier transform infrared spectroscopy (FTIR-ATR) was performed on a Thermo Nicolet NEXUS FTIR spectrometer (Thermo Scientific, Waltham, MA, USA) in absorption mode using 10 scans. The spectra were collected in the 600–4000 cm^−1^ range and the resolution was 2 cm^−1^.

Thermal properties were determined by differential scanning calorimetry (DSC), and analysis was conducted on a Mettler Toledo DSC 2010. Thermal stability was determined by thermogravimetric analysis (TG/TGA) using a TGA 2050 analyzer (TA Instruments, New Castle, DE, USA).

The molecular weight and molecular weight distribution of polymers were determined using a gel permeation chromatography system with a multiangle laser light scattering detector (GPC MALLS, DAWN HELEOS WYATT Technologies, Santa Barbara, CA, USA) and a refractive index detector (WGE Dr Bures GmbH & Co., KG, Dallgow-Döberitz, Germany, Dn-2010).

Scanning electron microscope (SEM) imaging was performed using a Hitachi TM 3000 electron microscope (Hitachi High-Technologies Corporation, Tokyo, Japan). Samples were mounted on aluminum stubs and coated with silver using standard sputtering techniques. The accelerating voltage used ranged from 5 to 15 kV.

## 4. Conclusions

A new class of vanadium(III) and nickel(II) catalysts bearing *N*-heterocyclic carbene (NHC) ligands was synthesized and evaluated for the polymerization of norbornene. The vanadium(III) complexes, activated with diethylaluminum chloride (AlEt_2_Cl), demonstrated good activity in the ring-opening metathesis polymerization (ROMP). In contrast, the nickel(II) complexes, activated with methylaluminoxane (MMAO), exhibited good activity in vinyl-addition polymerization of norbornene. The polymerization results reveal a clear distinction in catalytic efficiency between vanadium(III) and nickel(II) catalysts. Vanadium(III) catalysts (**3a** and **3b**) achieved significantly higher yields compared to nickel(II) catalysts (**4a** and **4b**), which produced lower yields under the same conditions. This outcome indicates that vanadium(III) catalysts exhibit a faster polymerization rate or higher initiation efficiency within the 2 h reaction time. The difference in yields can be attributed to the nature of the active species generated by the two catalyst systems. The V/AlEt_2_Cl/ETA system appears to produce highly reactive catalytic centers capable of rapid monomer conversion. In contrast, the Ni/MMAO system demonstrates slower catalytic turnover, likely due to differences in activation mechanisms, propagation rates, or stability of the active species. Despite the lower yield, nickel catalysts produce polymers with significantly higher molecular weights, suggesting that they promote slower chain initiation but longer chain growth per active site. This trade-off between high-yield V(III) catalysts and high-molecular weight Ni(II) catalysts reflects the fundamentally different polymerization dynamics inherent to the two metal centers and their respective cocatalysts. The catalytic activities are attributed to the steric and electronic properties provided by the NHC ligand. The structural characterization of the resulting polynorbornenes by NMR and FTIR confirmed the formation of both vinyl-type and ROMP-type polynorbornenes. Vinyl-type polynorbornenes showed significantly higher molecular weights than those obtained via ROMP. Notably, variations in the linker length between imidazole moieties in the nickel(II) catalysts led to distinct differences in polymer molecular weight, with a longer linker yielding higher molecular weight and increasing the homogeneity of the polymer. SEM imaging further demonstrated that linker length significantly influences polymer morphology, with a longer linker promoting a more fibrous and cohesive structure. These findings underscore the critical role of NHC ligand architecture, particularly steric bulk and linker length, on the activity, polymerization pathway, and the molecular characteristics of the resulting polynorbornenes.

The presented study shows the influence of NHC ligands on the efficiency of two metal centers, vanadium(III) and nickel(II), which act as catalysts via two different polymerization mechanisms in the homopolymerization of polynorbornene. Therefore, the influence of NHC ligands alone can be estimated. In a broader perspective, the research fits into the latest trends concerning nickel and vanadyl compounds [60,61], and therefore, in our opinion, may have more general application in many other fields.

## Figures and Tables

**Figure 1 ijms-26-06691-f001:**
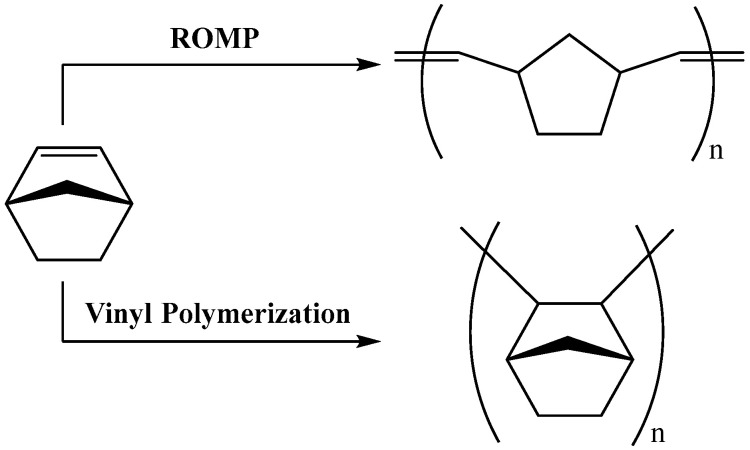
Schematic representation of the two types of coordination polymerization of norbornene (ROMP = ring-opening metathesis polymerization).

**Figure 2 ijms-26-06691-f002:**
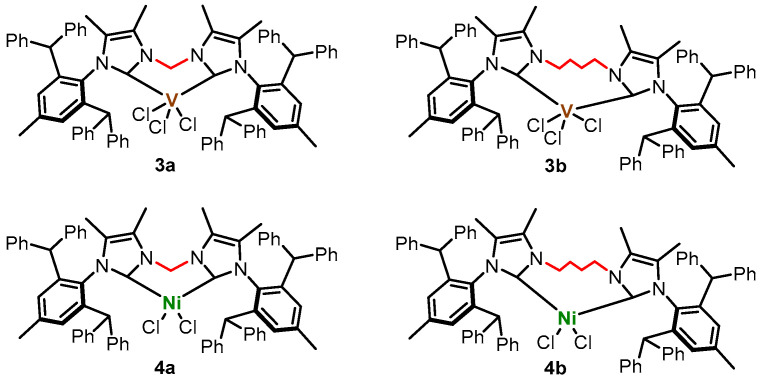
Structures of the catalysts used in this work.

**Figure 3 ijms-26-06691-f003:**
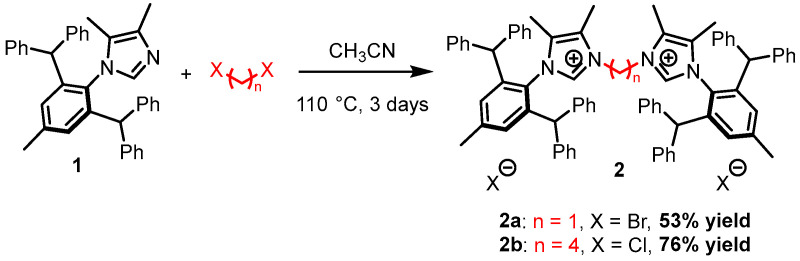
Synthesis of pre-ligands **2a** and **2b**.

**Figure 4 ijms-26-06691-f004:**
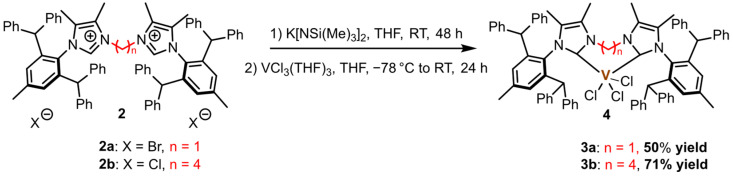
Synthesis of vanadium(III) complexes **3a** and **3b**.

**Figure 5 ijms-26-06691-f005:**
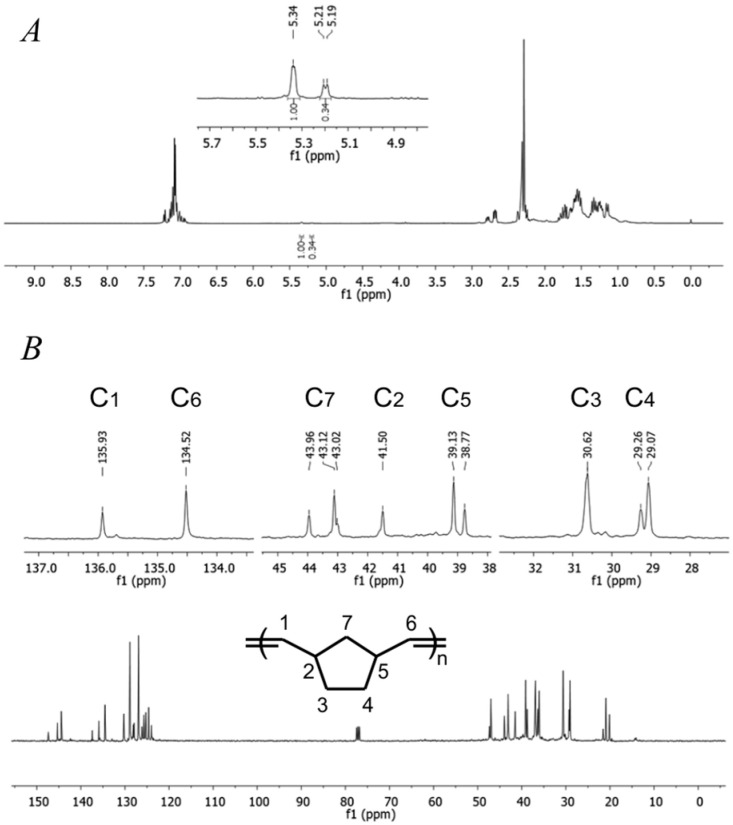
NMR spectra of the polynorbornene synthetized with vanadium(III) catalyst **3b**: (**A**) ^1^H NMR; (**B**) ^13^C NMR (CDCl3, 23 °C, TMS, internal standard).

**Figure 6 ijms-26-06691-f006:**
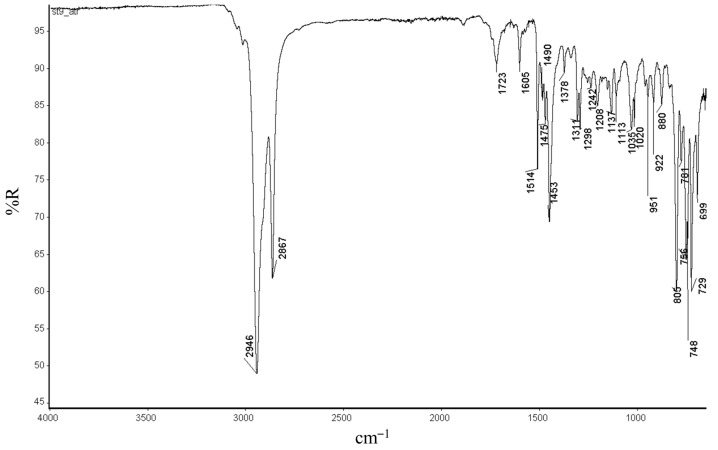
FTIR-ATR spectra of the polynorbornene synthetized with vanadium(III) catalyst **3b**.

**Figure 7 ijms-26-06691-f007:**
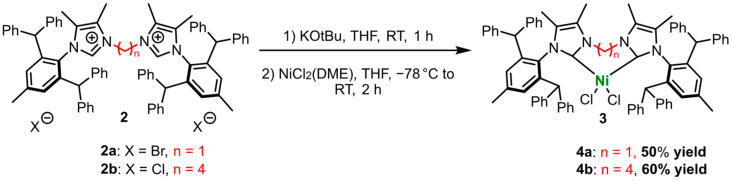
Synthesis of nickel(II) complexes **3a** and **3b**.

**Figure 8 ijms-26-06691-f008:**
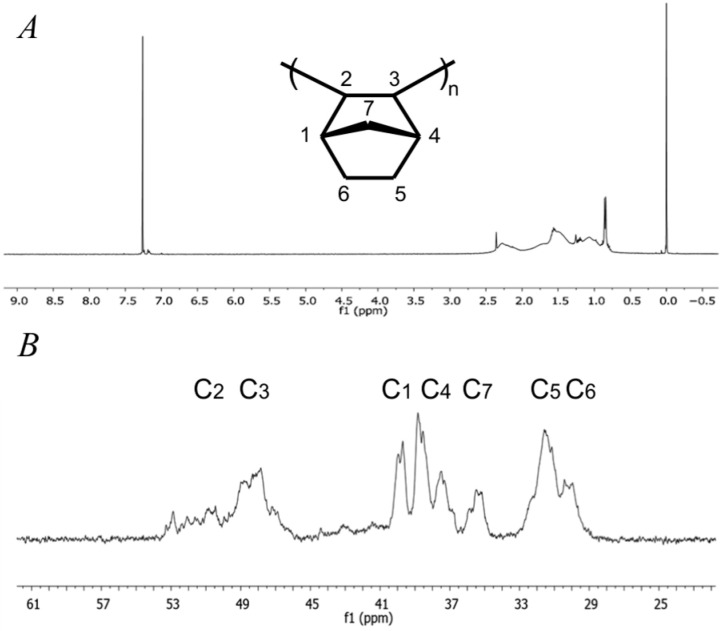
NMR spectra of the polynorbornene synthetized with nickel(II) catalyst **4b**: (**A**) ^1^H NMR; (**B**) ^13^C NMR (CDCl_3_, 23 °C, TMS, internal standard).

**Figure 9 ijms-26-06691-f009:**
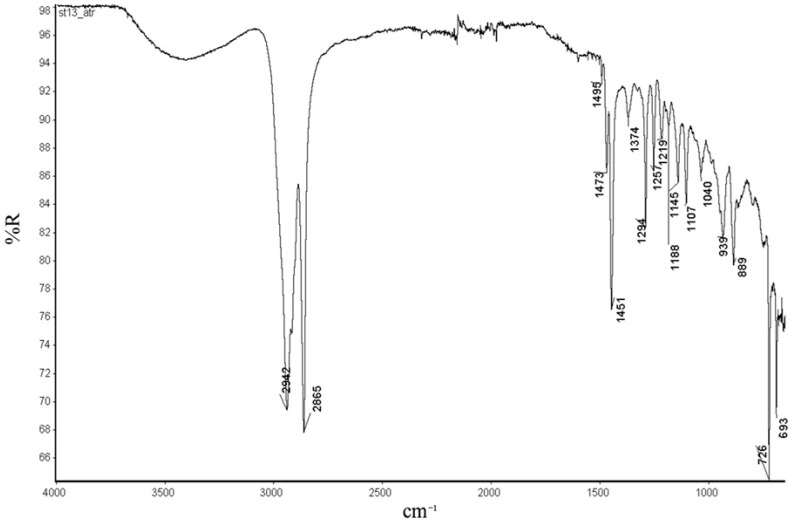
FTIR-ATR spectra of the polynorbornene synthetized with nickel(II) catalyst **4b**.

**Figure 10 ijms-26-06691-f010:**
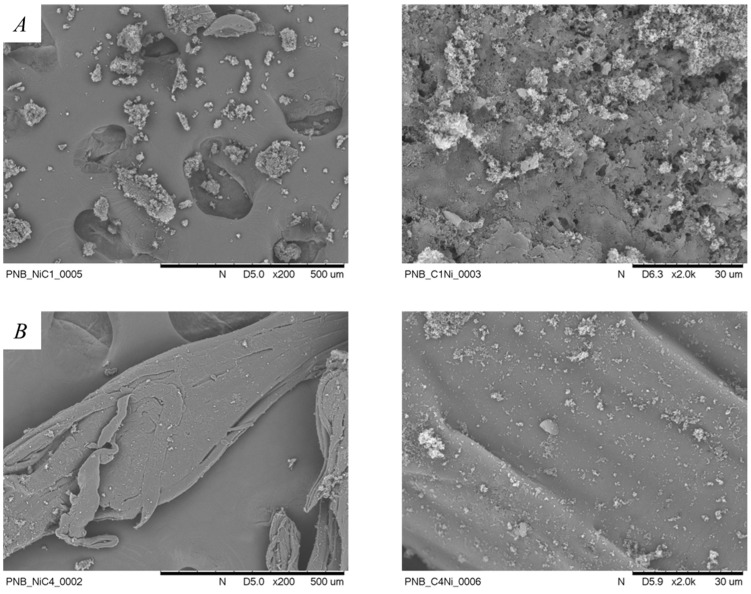
SEM images of (**A**) **4a**-derived polynorbornene and (**B**) **4b**-derived polynorbornene at ×200 and 2000 magnifications.

**Table 1 ijms-26-06691-t001:** Polymerization of norbornene by the vanadium(III) catalysts **3a** and **3b** and the nickel(II) catalysts **4b** and **4b** ^a^.

Catalyst	Cocatalyst	Yield [g]	Activity × 10^−3^ [g_PNB_/mol_Mt_/h]	M_w_ [kDa]	M_w_/M_n_
**3a**	AlEt_2_Cl, ETA	1.76	293.3	2.03	1.3
**3b**	AlEt_2_Cl, ETA	1.86	310.0	2.10	1.3
**4a**	MMAO	0.55	91.7	73.14	2.4
**4b**	MMAO	0.39	65.0	97.23	1.9

^a^ Polymerization conditions: vanadium(III) catalysts **3a**-**3b** or nickel(II) catalysts **4a**-**4b** = 3 × 10^−6^ mol; solvent—toluene; reaction temperature = 25 °C; Ni:MMAO = 1:1000; V:AlEt_2_Cl = 1:3000; 2.2 × 10^−4^ mol ETA; reaction time = 2 h.

## Data Availability

The original contributions presented in this study are included in the article/Appendix A. Further inquiries can be directed to the corresponding author.

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
