# Peer review of "ROMP and Vinyl Polynorbornenes with Vanadium(III) and Nickel(II) diNHC Complexes"

_ijms, 2025, doi:10.3390/ijms26146691_

Round 1
Reviewer 1 Report
Comments and Suggestions for Authors
The manuscript titled “ROMP and Vinyl Polynorbornenes with Vanadium(III) and Nickel(II) diNHC Complexes” by Halikowska-Tarasek, K.; et al. is a scientific work where the authors designed a chemical route to synthesize vanadium(III) and nickel(II) by ring openingmetathesis polymerization. The reaction steps were followed by NMR and FTIR measurements and the polymer surface morphology characterized by SEM. This is a topic of growing interest and furthermore, the manuscript is generally well-written.
However, it exists some points that need to be addressed (please, see them below detailed point-by-point) to improve the scientific quality of the submitted manuscript paper before this article will be consider for its publication in the International Journal of Molecular Sciences.
1) Introduction. “The structure and characteristics of polynorbornene (PNB) synthesized (…)” (lines 28-29). Could the authors provide quantitative data insights according to the existing Industry manufacturing applications related to polynorbornene and the properties of this polymer? This will significantly aid the potential readers to better understand the significance of this devoted research.
2) “Norbornene can undergo polymerization through two fundamental pathways, determined by the used catalysist: ring-opening metathesis polymerization (ROMP) and vinyl polymerization” (lines 29-32). What is the reaction yields related to each pathways? Some insights should be furnished in this regard.
3) Results. “The resulting nickel(II) complexes 4a and 4b were obtained as light green solids, with good yields of 50% and 60%, respectively” (lines 159-160). Did the authors notice an inhibition of reaction kinetic rates based on competitions effects coming from the side-reaction products? Some insights should be furnished in this regard.
4) “Moreover, scanning electron microscope (…) revealed a strong influence on polymer morphology” (lines 221-223). Did the authors visualize any cracking effect on the polymer architecture by SEM imaging? What was the surface roughness?
5) Finally, it may be also desirable to remark some future trends of nickel and vanadyl compounds like their integration on superconducing resonators [1] enhancing their quantum response by the optimization of the complex electron valency [2]. This will strengthen the importance of this research in many other fields.
[1] https://doi.org/10.1021/acs.jpcc.4c07265
[2] https://doi.org/10.1021/acs.inorgchem.2c03381
6) “4. Conclusions” (lines 263-281). This section perfectly remarks the most relevant outcomes found by the authors in this work. It may be desirable to add a brief statement to discuss about the potential future action lines to pursue the topic covered in this research and also the promising future perspectives.
Author Response
First of all, we would like to thank the reviewers for their valuable comments and time, which allowed us to improve the manuscript we submitted.
Comment 1: Introduction. “The structure and characteristics of polynorbornene (PNB) synthesized (…)” (lines 28-29). Could the authors provide quantitative data insights according to the existing Industry manufacturing applications related to polynorbornene and the properties of this polymer? This will significantly aid the potential readers to better understand the significance of this devoted research.
Response 1: Thank you for this comment. The introduction part was change, as below:
“… Polymers synthesized via the ROMP mechanism are unsaturated and typically demonstrate good solubility in a range of solvents and a low glass transition tempera-ture (Tg ≈ 35°C). The flexible backbone with reactive double bonds causes thermal and oxidative instability. On the other hand, reactive double bonds allow for a variety of functionalization, making poly(norbornene) ROMP suitable for optical components, biomedical scaffolds, hydrogels, drug delivery systems, and stimuli-responsive materi-als [4-12]. In contrast, the vinyl polymerization of norbornene produces saturated, 2,3-inserted polymers, which possess unique chemical and physical properties, such as high thermal stability, and high glass transition temperature values ranging from 180 to 370 °C, depending on the functional groups. It is amorphous, optically transparent, chemically resistant, and has low moisture absorption, making it ideal for high-performance applications such as microelectronic dielectrics, optical films, and LCD cover layers [1-3, 12-21]. …”
Additionally, new references have been added (Ref. 1 and Ref. 3), which required renumbering the citations throughout the manuscript.
- Park, K. H.; Twieg, R. J.; Ravikiran, R.; Rhodes, L. F.; Shick, R. A.; Yankelevich, D.; Knoesen, A. Synthesis and Nonlinear-Optical Properties of Vinyl-Addition Poly(norbornene)s. 2004, 37, 5163-5178.
- Blank, F.; Janiak, C. Metal catalysts for the vinyl/addition polymerization of norbornene. Chem. Rev. 2009, 253, 827–861.
- Wang, X.; Jeong, Y. L.; Love, C.; Stretz, H. A.; Stein, G. E.; Long, B. K. Design, synthesis, and characterization of vinyl-addition polynorbornenes with tunable thermal properties, Chem. 2021, 12, 5831-5841.
Comment 2: “Norbornene can undergo polymerization through two fundamental pathways, determined by the used catalyst: ring-opening metathesis polymerization (ROMP) and vinyl polymerization” (lines 29-32). What is the reaction yields related to each pathways? Some insights should be furnished in this regard.
Response 2: Thank you for the valuable comment. In response, we have added to the manuscript a detailed statement to the conclusion section. “… The polymerization results reveal a clear distinction in catalytic efficiency between vanadium(III) and nickel(II) catalysts. Vanadium(III) catalysts (3a and 3b) achieved significantly higher yields compared to nickel(II) catalysts (4a and 4b), which produced lower yields under the same conditions. This outcome indicates that vanadium(III) catalysts exhibit a faster polymerization rate or higher initiation efficiency within the 2 hour reaction time. The difference in yields can be attributed to the nature of the active species generated by the two catalyst systems. The V/AlEtâ‚‚Cl/ETA system appears to produce highly reactive catalytic centers capable of rapid monomer consumption. In contrast, the Ni/MMAO system demonstrates slower catalytic turnover, likely due to differences in activation mechanisms, propagation rates, or stability of the active species. Despite the lower yield, nickel catalysts produce polymers with significantly higher molecular weights, suggesting that they promote slower chain initiation but longer chain growth per active site. This trade-off between high yield (V(III)-catalysts) and high molecular weight (Ni(II)-catalysts) reflects fundamentally different polymerization dynamics inherent to the two metal centers and their respective cocatalysts. …”
Comment 3: “The resulting nickel(II) complexes 4a and 4b were obtained as light green solids, with good yields of 50% and 60%, respectively” (lines 159-160). Did the authors notice an inhibition of reaction kinetic rates based on competitions effects coming from the side-reaction products? Some insights should be furnished in this regard.
Response 3: The inhibition of the kinetic reaction rates has not been studied. Usually, the kinetic profile of the reaction is determined using a metal complex as a catalyst (e.g. DOI: 10.1039/d4cy01315h). However, it is not determined during the synthesis of the complexes. The efficiency of the synthesis is generally dependent on the type of metal and ligands. In our laboratory, the synthesis of complexes usually results in higher yields (e.g. https://doi.org/10.1021/acs.organomet.2c00262). In the present work, a similar synthetic art was used, and in our opinion, the obtained efficiency can be considered as good.
Comment 4: “Moreover, scanning electron microscope (…) revealed a strong influence on polymer morphology” (lines 221-223). Did the authors visualize any cracking effect on the polymer architecture by SEM imaging? What was the surface roughness?
Response 4: It is a valuable remark, thank you. The morphology of both polymers was significantly different. The polymer obtained over the nickel catalyst (4a) with the pre-ligand 2a (C1 bridge) had a rough, porous surface morphology. In contrast, the PNB obtained over the nickel catalyst with the pro-ligand 2b (C4 bridge) had a smooth surface without visible defects. No evidence of cracking or fracture was observed in the SEM images under the conditions studied, indicating that the polymer morphology remains intact without damage or defects. To better show the surface of the polymers, additional SEM images of the PNB at × 2000 magnification have been added to the manuscript (Figure 10).
Comment 5: Finally, it may also be desirable to remark some future trends of nickel and vanadyl compounds like their integration on superconducing resonators [1] enhancing their quantum response by the optimization of the complex electron valency [2]. This will strengthen the importance of this research in many other fields.
Response 5: We agree. We have found both references suitable in a broader perspective of the application of nickel and vanadium compounds, and therefore, they were implemented in the main text.
Comment 6: “4. Conclusions” (lines 263-281). This section perfectly remarks the most relevant outcomes found by the authors in this work. It may be desirable to add a brief statement to discuss about the potential future action lines to pursue the topic covered in this research and also the promising future perspectives.
Response 6: Thank you for this comment. In response, we have added a brief statement of a broad perspective at the end of the conclusion section. Once again, we would like to thank you very much for all your comments and your time.
Reviewer 2 Report
Comments and Suggestions for Authors
This manuscript aims to describe the synthesis and catalytic application of ROMP and Vinyl Polynorbornenes with Vanadium(III) and Nickel(II) diNHC Complexes. Although the topic is interesting in its scientific field, there are some issues that require the authors’ attention to improve the quality of this particular manuscript before further consideration for publication in a high-quality journal “IJMS”.
Specific comments:
- The authors should clearly state the novelty of this type of bidentate NHC ligand in V(III)/Ni(II)-catalyzed ROMP.
- Why the authors simultaneously used 1H and 13C NMR to identify the structures of ROMP-type and vinyl-type PNB? Please specify.
- Although Table 1 provided the yields and activities of ROMP and vinyl catalytic reactions, there is a lack of in-depth discussion on the differences in catalytic activity.
- This study only listed Tg and Tm values. Please provide the data regarding thermal stability and mechanical properties.
- As mentioned in the Section 3.2, 1H NMR and 13C NMR spectra were recorded on Bruker spectrometer at 400 (1H 245 NMR) and 100 MHz (13C NMR). Nevertheless, in my opinion, this important experimental claim involving simultaneous use of proton and carbon NMR is not supported by any appropriate documentation. If possible, please consider the inclusion of the following relevant case study (DOI: 10.1016/j.mtbio.2021.100183) in the reference list to strengthen manuscript quality and attract more attention from broad readers.
Author Response
First of all, we would like to thank the reviewers for their valuable comments and time, which allowed us to improve the manuscript we submitted.
Comment 1: The authors should clearly state the novelty of this type of bidentate NHC ligand in V(III)/Ni(II)-catalyzed ROMP.
Response 1: Thank you for this comment. To the best of our knowledge, this study represents the first application of solely bidentate NHC ligands in V(III)- and Ni(II)-catalysed polymerisation of norbornene. Moreover, there are very few reported examples of bis-NHC ligands being employed in polymerisation catalysis in general [42-46]. Also, vanadium catalysts are not much explored for norbornene polymerisation. To respond to this comment, the introduction has been rewritten. The novelty of the work has been emphasised.
Comment 2: Why the authors simultaneously used 1H and 13C NMR to identify the structures of ROMP-type and vinyl-type PNB? Please specify.
Response 2: Thank you for this comment. Both ¹H and ¹³C NMR spectroscopy are used to comprehensively confirm the structures of ROMP-type and vinyl-type polynorbornenes because each method provides complementary information. ¹H NMR clearly allows detection of backbone double bonds methine protons, which are present in ROMP-derived PNB but absent in vinyl-type PNB. The presence of vinylic proton signals is a clear marker distinguishing ROMP products from saturated vinyl-addition polymers. ¹³C NMR provides detailed information about the carbon skeleton, confirming the saturation or unsaturation of the polymer backbone. In vinyl-type PNB, ¹³C NMR confirms the fully saturated backbone carbons, while in ROMP-type PNB, it identifies the characteristic olefinic carbons from the unsaturated backbone. Using both techniques ensures accurate differentiation between the two polymer types, confirms polymerisation pathways, and verifies the structural integrity of the products.
No less important is the fact that one of the polymers was obtained in the form of oil. In such a case, an unambiguous determination of the structure in the presence of signals from the solvents is important.
Finally, if we have some general data (like both 1H and 13C NMR spectra) it would be worth sharing.
Comment 3: Although Table 1 provided the yields and activities of ROMP and vinyl catalytic reactions, there is a lack of in-depth discussion on the differences in catalytic activity.
Response 3: Thank you for the valuable comment. In response, we have added to the manuscript a detailed statement in the conclusion section. “… The polymerisation results reveal a clear distinction in catalytic efficiency between vanadium(III) and nickel(II) catalysts. Vanadium(III) catalysts (3a and 3b) achieved significantly higher yields compared to nickel(II) catalysts (4a and 4b), which produced lower yields under the same conditions. This outcome indicates that vanadium(III) catalysts exhibit a faster polymerisation rate or higher initiation efficiency within the 2-hour reaction time. The difference in yields can be attributed to the nature of the active species generated by the two catalyst systems. The V/AlEtâ‚‚Cl/ETA system appears to produce highly reactive catalytic centers capable of rapid monomer consumption. In contrast, the Ni/MMAO system demonstrates slower catalytic turnover, likely due to differences in activation mechanisms, propagation rates, or stability of the active species. Despite the lower yield, nickel catalysts produce polymers with significantly higher molecular weights, suggesting that they promote slower chain initiation but longer chain growth per active site. This trade-off between high yield (V(III)-catalysts) and high molecular weight (Ni(II)-catalysts) reflects fundamentally different polymerization dynamics inherent to the two metal centers and their respective cocatalysts. …”
Comment 4: This study only listed Tg and Tm values. Please provide the data regarding thermal stability and mechanical properties.
Response 4: Thank you for the valuable comment. PNB obtained using the vanadium catalysts has a form of oil (Supporting Information), therefore, we do not investigate its properties in a more detailed way. In contrast, the PNB obtained using the nickel catalysts has a form of white solid powder (Supporting Information), and turned out to be thermally very stable. In particular, PNB obtained with the nickel catalyst with the ligand having C4 bridge (4b). Therefore, in the Supporting Information, the TGA curves of the PNB obtained using the nickel catalyst were added.
Comment 5: As mentioned in the Section 3.2, 1H NMR and 13C NMR spectra were recorded on Bruker spectrometer at 400 (1H 245 NMR) and 100 MHz (13C NMR). Nevertheless, in my opinion, this important experimental claim involving simultaneous use of proton and carbon NMR is not supported by any appropriate documentation. If possible, please consider the inclusion of the following relevant case study (DOI: 10.1016/j.mtbio.2021.100183) in the reference list to strengthen manuscript quality and attract more attention from broad readers.
Response 5: Thank you for this comment. Indeed, the proposed reference (below) includes both 1H and 13C NMR analysis. Since this reference also covers polymer analysis, we considered it valuable and therefore added it to the main text in the appropriate section. Reference: Nguyen, D. D.; Luo L.-J.; Lai J.-Y. Thermogels containing sulfated hyaluronan as novel topical therapeutics for treatment of ocular surface inflammation. Materials Today Bio 2022, 13, 100183. https://doi.org/10.1016/j.mtbio.2021.100183
Once again, we would like to thank you very much for all your comments and your time.
Round 2
Reviewer 2 Report
Comments and Suggestions for Authors
The revised version has adequately addressed most of the critiques raised by this reviewer and is now suitable for publication in "IJMS".